# Microglia and Inhibitory Circuitry in the Medullary Dorsal Horn: Laminar and Time-Dependent Changes in a Trigeminal Model of Neuropathic Pain

**DOI:** 10.3390/ijms22094564

**Published:** 2021-04-27

**Authors:** Nuria García-Magro, Yasmina B. Martin, Pilar Negredo, Francisco Zafra, Carlos Avendaño

**Affiliations:** 1Department of Anatomy, Histology and Neuroscience, Medical School, Autónoma University of Madrid, 28029 Madrid, Spain; nuria.garciamagro@gmail.com (N.G.-M.); pilar.negredo@uam.es (P.N.); 2Ph.D. Programme in Neuroscience, Doctoral School, Autónoma University of Madrid, 28049 Madrid, Spain; 3Departamento de Anatomía, Facultad de Medicina, Universidad Francisco de Vitoria, Pozuelo de Alarcón, 28223 Madrid, Spain; y.martin.prof@ufv.es; 4Centro de Biología Molecular Severo Ochoa, Consejo Superior de Investigaciones Científicas, Universidad Autónoma de Madrid, 28049 Madrid, Spain; fzafra@cbm.csic.es

**Keywords:** chronic pain, allodynia, trigeminocervical complex, glycine transporters

## Abstract

Craniofacial neuropathic pain affects millions of people worldwide and is often difficult to treat. Two key mechanisms underlying this condition are a loss of the negative control exerted by inhibitory interneurons and an early microglial reaction. Basic features of these mechanisms, however, are still poorly understood. Using the chronic constriction injury of the infraorbital nerve (CCI-IoN) model of neuropathic pain in mice, we have examined the changes in the expression of GAD, the synthetic enzyme of GABA, and GlyT2, the membrane transporter of glycine, as well as the microgliosis that occur at early (5 days) and late (21 days) stages post-CCI in the medullary and upper spinal dorsal horn. Our results show that CCI-IoN induces a down-regulation of GAD at both postinjury survival times, uniformly across the superficial laminae. The expression of GlyT2 showed a more discrete and heterogeneous reduction due to the basal presence in lamina III of ‘patches’ of higher expression, interspersed within a less immunoreactive ‘matrix’, which showed a more substantial reduction in the expression of GlyT2. These patches coincided with foci lacking any perceptible microglial reaction, which stood out against a more diffuse area of strong microgliosis. These findings may provide clues to better understand the neural mechanisms underlying allodynia in neuropathic pain syndromes.

## 1. Introduction

Neuropathic pain in the craniofacial territory is associated with different pathologies, including trigeminal neuralgia, nerve entrapments and chemical, tumoral or viral neuropathies, which most often cause chronic and debilitating painful conditions that affect millions of people worldwide and are difficult to treat (International Classification of Headache Disorders 2018). Although still far from complete, the understanding of its complex pathophysiology has significantly advanced in recent years thanks to an effective dialogue between basic and clinical efforts, which is being instrumental to develop new and more effective therapies. As well as for other body regions, two key mechanisms underlie this kind of pain. Firstly, it is accompanied by a loss of the negative control that inhibitory interneurons normally exert on the transmission of nociceptive and non-nociceptive signals toward ascending pathways [1,2,3]. Consistently, the pharmacological blockade of this inhibition results in symptoms that resemble those present in neuropathic pain patients [4,5], whereas the administration of GABA-enhancing drugs or vectors, or GABA or Glycine transporter inhibitors reverse or block nerve injury-induced pain behavior [6,7,8,9,10]. And secondly, nerve injury provokes a swift microglial reaction in the terminal fields of primary afferents in the superficial laminae of the dorsal horn [11]. This reactive microglia releases BDNF, activating its high-affinity receptor TrkB, which enhances pain transmission by projection neurons in the superficial laminae of the dorsal horn through a variety of mechanisms [12,13,14,15,16,17].

Among the animal models that mimic chronic pain in the orofacial region, the chronic constriction injury of the infraorbital nerve (CCI-IoN) is probably the most widely used [18,19]. Neuropathic pain mechanisms in the trigeminal territory resemble those in spinal territories, but only to a degree, because of differences in the cellular composition and circuitries, cellular genomic signatures, the range of painful conditions specific to the face and head, and differential sensitivities to some pharmacological treatments [20,21,22,23,24]. Also, the few studies that report on the microgliosis that follows CCI-IoN, provide evidence of some differences between reactive microglia in trigeminal and spinal territories [25,26]. All in all, many cellular and molecular features of these mechanisms, however, are still incompletely defined. The present report was therefore aimed at further clarifying what spatial and temporal correlations, if any, exist between the reactive microgliosis and the expression of key molecules involved in the inhibitory regulation of neural transmission in the medullary dorsal horn in a model of trigeminal neuropathic pain.

## 2. Results

### 2.1. Behavior

Mice in the CCI-IoN group developed significant allodynia to mechanical stimulation with Von Frey filaments of either strength or air pulses by 14 postoperative days (dpo), and this increased at 21 dpo (Figure 1). On the contralateral side the response score increased in a few cases, without reaching statistical differences with controls (data not shown).

### 2.2. GAD-6 and GlyT2 Immunostaining

In controls, GAD-6 was strongly expressed as a homogeneous band in laminae I-II, and more moderately in laminae III-IV. GlyT2, in contrast, predominated in the inner part of lamina II and in irregular ‘patches’ within lamina III. The outer part of lamina II and laminae III and IV outside the ‘patches’ exhibited lower and more variable expression of this transporter (Figure 2). Neither GAD-6 nor GlyT2 significantly labeled cell bodies, confirming their affinity for axons and terminals [27,28].

Following CCI-IoN, GAD-6 expression significantly decreased in all laminae of the ipsilateral side by 5 dpo, and this decline continued at least until 21 dpo (Figure 2 and Figure 3). A parallel, but minor decrease was also observed in lamina III of the contralateral side compared to controls (Figure 3A). The frequency distribution curves of pixel intensities showed global reductions of immunofluorescence intensity (Figure 3A), and these reductions reached statistical significance, particularly at 21 dpo (Figure 4). The expression of GlyT-2 was also affected by CCI-IoN, but in a more complex manner (Figure 3B): at 5 dpo, GlyT-2 expression decreased only in lamina III, compared to controls and to the side contralateral to CCI-IoN; at 21 dpo, however, this reduction was significant in all laminae. The frequency distribution curves of pixel intensities, however, overlapped substantially, resulting in no statistical differences. Also, as will be described below, changes in GlyT2 expression varied in different domains of the spinal trigeminal nucleus (Sp5C).

### 2.3. Microglia

Iba-1 immunolabeled microglial cells distributed homogeneously, with relatively low density, across all layers of Sp5C in controls. Morphologically, these cells fit the description of ‘resting’ or ‘surveying’ microglia (Figure 5A,D; see [29]). Five days after CCI, a strong microgliosis was observed, consisting of a notable increase in cell number and intricacy of microglial processes. This reaction was more prominent in laminae I-II, and more moderate in laminae III-IV, and persisted, although in a diminished form, at 21 dpo (Figure 5B,C,E).

Densitometric measurements showed that the increased expression of Iba-1 in the side ipsilateral to the CCI differed significantly from both the contralateral side of the same cases, and the control cases. This difference decreased, but remained significant, at 21dpo (Figure 6).

### 2.4. Microgliosis and GlyT2 Expression

The microgliosis induced in the Sp5C by the CCI-IoN did not occur homogeneously across the nucleus. Rather, there was an unexpected spatial restriction of the reactive microglia to the sectors of the nucleus where the expression of GlyT2 was weaker, thus sparing the ‘patches’ that displayed a stronger immunofluorescence for the transporter (Figure 7). This phenomenon was consistently observed in the three cases where it was investigated (Figure 8).

## 3. Discussion

The general aim of the present study was to investigate well-defined molecular and cellular events underlying the general process of disinhibition that results from partial peripheral nerve injury, in a model of facial neuropathic pain. Applying CCI-IoN on mice, this study examines the evolution of the changes in expression of GAD, as a marker of GABAergic cells, GlyT2, as a marker of glycinergic axon terminals, and microglia, the glial cells that more rapidly respond to the nerve injury. The following major findings emerged: (1) Our results confirm that the constriction of the IoN reduces the expression of GAD and GlyT2, key markers of GABAergic and Glycinergic inhibitory mechanisms respectively, in the medullary dorsal horn. (2) Both markers differ in their laminar distribution pattern and temporal course of change after nerve injury. GAD expression predominates in laminae I-II, being relatively uniform across each lamina, decreases markedly 5 days after CCI-IoN, and decreases further by 21 days. GlyT2 expression, on the other hand, predominates in lamina III where it shows irregular ‘patches’ of strong expression within a ‘matrix’ of weaker expression. After CCI-IoN, GlyT2 decreases more markedly at 21 days postinjury, and in the GlyT2-poorer ‘matrix’. (3) CCI-Ion induces a dramatic microgliosis 5 days after lesion, which recedes substantially by 21 days, but still maintaining significantly higher levels against the contralateral side or naïve controls. (4) The simultaneous analysis of GlyT2 expression and microglial activation documents an unexpected spatial restriction of the reactive microglia to the sectors of the nucleus where the expression of GlyT2 was weaker, a heretofore undescribed relationship between these key elements of neuropathic pain. 

### 3.1. Inhibitory Transmitters in Sp5C Are Affected by the CCI-IoN

Synaptic inhibition in the spinal or medullary dorsal horn has long been postulated [30] and eventually shown [31,32,33,34,35,36] to control the transmission of nociceptive impulses to higher brain centers. Key players in this inhibition are the local GABAergic and Glycinergic interneurons, which constitute one-third of all neurons in laminae I-IV of the Sp5C [37]. Peripheral nerve injury drives a loss of GABA and Glycinergic inhibition in both the spinal [1,4,5,31,35,38,39,40,41] and the medullary [2,42,43] dorsal horn.

Early reports suggested that the number of GABA-immunoreactive cells in the spinal dorsal horn started to decline by 2 weeks after a nerve transection and continued to reach a 30% loss by 4 weeks postsurgery [38]. GABA- or Glycine-immunoreactive neuron loss was later ruled out, at least 2 weeks after CCI of the sciatic nerve [44], therefore pointing to other mechanisms as more probable explanations of the nerve injury-induced neuropathic pain, such as decreased transmitter release or activity of these neurons, reduced capacity of GABA and glycine to generate inhibition [1,45,46], or potentiation of nearby postsynaptic NMDA receptors by ‘spillover’ of unrecaptured glycine at the synapse [47]. Our results support that CCI-IoN is followed by decreased GABAergic and Glycinergic neurotransmission, as revealed by reductions in the expression of GAD and GlyT2 respectively, which display singular spatial and temporal features.

As previously reported, GAD expression is strong in laminae I and II, and more moderate in deeper layers, without major irregularities across the mediolateral extent of each lamina [48,49,50]. The reduction of GAD (or GABA) expression triggered by partial peripheral nerve injuries is, accordingly, more pronounced in superficial laminae, and remains confined to the mediolateral sector innervated by the affected nerve [2,48,50]. Compared with our previous data [2] present results show a larger mediolateral extent of GAD reduction, which may be due to the slanted projection of the IoN on the caudal trigeminal nuclei, which invades more medial sectors at rostral levels of Sp5C, and recedes towards more lateral sectors at more caudal levels and upper cervical segments [51]. A similar topography was found at the same levels for GlyT2. The contralateral reduction of GAD in lamina III is compatible with the existence of a small contingent of crossed primary trigeminal afferents terminating in laminae III-V [52].

In contrast with the well agreed upon distribution of GAD and GABA in the spinal and medullary dorsal horn, the irregularities we observed in the expression of GlyT2 have not been reported before. Since the patch/matrix arrangement of this marker occurs both in naïve and in operated animals, it seems to reflect a constitutive property in the distribution of this transporter. Moreover, the antibody used against GlyT2 specifically stains synaptic boutons, and only weakly cell bodies and axons [28], which suggests that glycinergic innervation displays preferential domains on a ‘background’ of sparser, particularly in lamina III. Irregularities in GlyT2 distribution in the same lamina may be present as well in the spinal cord (see Figure 7 in [53]). It would be worth testing whether these domains match others of different cellular or molecular composition. Although most reports of the distribution and the expression of different molecules and cell populations fail to show obvious heterogeneities across each dorsal horn lamina, some exceptions may be found, such as the irregular grouping of the largely inhibitory parvalbumin-expressing interneurons [54] or of glycine receptors and gephyrin, a major scaffolding protein at inhibitory synapses [55,56].

### 3.2. Microglia Activation and Its Relationship with the Expression of Inhibitory Molecules

A strong association between reactive microglia and the GABA and Glycine inhibitory mechanisms is now established beyond doubt. CCI, as any other peripheral nerve injury, induces a rapid activation of microglia in the spinal or brain stem territories that received the affected primary afferents [43,57,58,59,60,61]. Although some authors proposed that neuropathic pain was primarily due to the downregulation of inhibitory transmission, with microglia activation being just a secondary phenomenon [40], other studies hold that nerve injury initiates a cascade of events involved in the development and maintenance of neuropathic pain [12,62,63]. Summarily (see [17,64], for recent reviews), microglia is activated by cytokines and chemokines released by injured peripheral nerve fibers, which increases the synthesis and membrane insertion of several purinergic receptors of the P2X ion-channel family. ATP released from primary afferents, the microglia itself, and local interneurons activates P2X4 receptors, resulting in the release within minutes of BDNF [65] which binds to its high-affinity receptor trkB. This occurs on lamina I nociceptive projection neurons, where it produces the downregulation of the neuronal potassium-chloride cotransporter KCC2, increasing the intracellular Cl^−^ and causing a reversal in the equilibrium potential of Cl^−^. The subsequent reduction of the inhibition mediated by GABAA receptors has been involved in both the induction and the maintenance of neuropathic pain [66]. Together with this postsynaptic effect, BDNF also acts presynaptically on peptidergic primary afferents of laminae I and II and, when combined with nerve injury and sustained KCC2-dependent disinhibition, changes the subunit composition of presynaptic NMDA receptors, potentiating transmission from these afferents [67,68,69,70].

The temporal course of these events offers clues to better understand the mechanisms underlying neuropathic pain. Early reports of a disinhibition of neurons in the sensory cortex, attributable to a loss of local GABAergic control, could indicate a different time-course of events. Such disinhibition happened immediately after peripheral nerve injuries, leading to an expansion of receptive fields in the cerebral cortex [71,72,73]. A similar effect was also found in the dorsal column nuclei after applying local anesthesia to a digit or toe [74,75]. Should a likewise rapid phenomenon take place following pain-inducing nerve lesions, our findings could suggest that changes in GABA are the earliest mechanistic explanation of the disinhibition observed. However, while disinhibition rapidly affecting the structure of receptive fields has been mostly attributed to a loss of lateral GABAergic inhibition [76], the implication of GABA in deafferentation-driven disinhibition leading to pain involves other mechanisms. These include GABA becoming less hyperpolarizing (or even depolarizing) because of a shift in the chloride equilibrium potential of some nociceptive dorsal horn neurons [13,53], and the disinhibition of a chain of excitatory feed-forward connections throughout superficial dorsal horn laminae towards nociceptive-projection neurons [77]. From our results, however, and in contrast with GAD, the expression of GlyT2 is only moderately decreased at 7 days post CCI-IoN, and this reduction grows in the following weeks. The relatively delayed downregulation of GlyT2 notwithstanding, glycine neurotransmission is a necessary condition for pain to appear, since this fails to occur when glycinergic cells are molecularly silenced without preventing microglial activation [40,78].

It seems, then, that microglia is better positioned to play a key role in the events that take place shortly after nerve injury. These cells already show morphological signs of activation 12 h after spinal nerve ligation, and proliferate between 32 h and 7 days postsurgery [79]. Moreover, they probably migrate soon to the affected region, strongly attracted by ATP and other chemoattractans that are promptly released by injured terminals in the dorsal horn [80]. By 24 h after the same manipulation or CCI of a peripheral nerve (before allodynia is detected), microglia activation was observed, as identified by Iba-1 or OX-42 increased immunoreactivity [26,81,82,83]. A prominent early effect of nerve ligation is the increase of BDNF in the dorsal horn, that reaches significance by 12 h after lesion, peaks one day later, and progressively disappears at 14 (as determined by immunohistochemistry) or 28 days (determined by ELISA; [67]). BDNF was then proposed to be associated only with the initiation of neuropathic pain, while the persistence of pain at later stages (beyond the second day postinjury) would depend on the activation of presynaptic NMDA receptors, particularly in primary afferents. In neuropathic, but not naïve rats, these receptors quickly change their composition to express the 2B subunit and upon activation stimulate glutamate release from those fibers [67,68,84]. The identification of early and late molecular responders in the pathophysiological mechanisms of neuropathic pain is also consistent with the existence of an early-responding microglia, which would be activated only within the first few days after injury, and a late-responding microglia, which would be more prevalent between the first and the fourth postinjury weeks. Although both populations would be morphologically indistinguishable, they showed markedly divergent transcriptomic profiles, with hundreds of genes being differentially regulated [85,86]. A cross-talk between microglia and astrocytes, whereby the chemokine CCL2 is positively regulated in astrocytes [87] contributes to the maintenance of microglial activation, which is particularly relevant for the chronification of neuropathic pain.

Early studies suggested that microglia activation did not last more than 2–4 weeks, reverting then to a ‘resting’ phenotype. Our results also indicate that the strong microgliosis found 7 days post CCI-IoN persists, but to a much milder degree, by 21 days postinjury. More recently, however, activated microglia has been demonstrated up to 98 days after irreversible and more damaging spinal or peripheral nerve pain-provoking injuries [88,89,90,91]. Interestingly, the time course of microglia activation parallels that of changes in the expression of key molecules involved in the signalling and regulation through activated microglia, such as BDNF, KCC2, GABA, GAD65 (but not GAD67), p38 mitogen-activated protein kinase, or transcription factors involved in the production of P2X4 receptors [1,17,67,92,93,94,95].

Finally, the intriguing finding of the patches of low microglial activation and high GlyT2 expression is difficult to explain. It is possible, nevertheless, that a variable topography and irregular degree of damage to the different fascicles of the IoN by the ligature result in uneven patterns of innervation by damaged afferents in the medullary dorsal horn. If so, foci of less affected innervation coinciding with areas of pre-existing stronger glycinergic innervation (the ‘patches’ in lamina III) would fail to create a sufficiently ‘attractive’ environment for microglia migration and activation. In any case, this is an unclarified line of investigation that would be worth pursuing.

## 4. Materials and Methods

### 4.1. Experimental Subjects

Fifteen young adult (2–3 months old) male C57BL/6 mice were used. They were group housed (4–6 animals/cage) in a 12 h light/dark cycle with access to food and water ad libitum. All protocols for animal procedures were approved in advance by the Ethical Committee of the Autónoma University of Madrid, in accordance with European Community’s Council Directive 2010/63/UE. The experimental design (Figure 9) was planned to minimize the suffering of the animals, and to keep their number to the minimum that was expected to provide reliable results.

### 4.2. Surgery and IoN Constriction

An infraorbital nerve constriction (CCI-IoN) was performed on the right side to induce a moderately painful condition resembling trigeminal neuralgia [18,96]). Under i.m. injection of Ketamine (Ketolar, 55 mg/kg, Pfizer; New York, NY, USA), Xylazine (Rompun, 15 mg/kg, Bayer; Leverkusen, Germany) and Atropine (0.2 g/kg), the IoN was exposed under the vibrissal pad and a single polypropylene monofilament (Surgipro 6.0, Covidien; Dublin, Ireland) ligature was loosely tied around the distal part of the nerve. As shown in rats, this procedure alters little, if at all, the circulation through the superficial epineural vasculature [2,97]). The skin wound was closed with interrupted silk sutures. Age-matched control mice were not operated. After recovery from surgery the operated animals’ motor, grooming or eating behavior did not differ from the controls. Because our previous experience in rats [2,43] and mice [29,96] showed no differences in mechanical responses to facial stimulation and microglial phenotype in the Sp5C between controls and sham-operated animals with simple facial skin incisions, no sham group was included.

### 4.3. Behavioral Testing

Testing for behavioral responses to mechanical stimulation of the whisker pad was performed according to the procedure described by Krzyzanowska et al. [96]. In short, the animals were habituated to the environment (a quiet room with red lighting) and the experimenter during 1 h daily. The experiments started the next day for three consecutive days before IoN surgery, to establish the baseline response score, and on postoperative days 5 and 21. The animals were partially restrained in a home-made device [96] that allowed them to expose and move freely the head and forelimbs, so that testing and response recording could easily be made, while producing low stress levels.

Testing started using, in this order, 0.008 g and 0.07 g von Frey hairs (North Coast Medical, Inc. Morgan Hill, CA, USA). Each hair was applied to the vibrissal pad until bent in 3 series of 5 pokes, starting randomly on the left or right sides, and changing sides after each series. The scores given to the responses followed previous description [96], ranging from 0.25 points to clear and brisk face withdrawal or aggression/biting of the probe to 1.5 points to repeated scratching of the face. For each animal, an overall “response score” was obtained by adding up the points separately for each testing time point. Following Von Frey testing, air-puff testing was done by delivering 20 ms pulses of air at 10 psi and 1 Hz through an 18 gauge metal tube with its tip perpendicular to, and about 1 cm distant from the whisker pad. Pulses were controlled by a pneumatic pump (Pico Spritzer II, General Valve Co., Fairfield, NJ, USA) activated by a pulse generator (Cibertec, Madrid, Spain). The same scoring criteria as in Von Frey testing were assigned to the responses observed in the animal.

### 4.4. Tissue Processing and Immunostaining

On the day after the last testing, the mice were deeply anesthetized (Dolethal, 50 mg/kg i.p., Vétoquinol; Madrid, Spain) and perfused through the ascending aorta with 0.9% NaCl (50 mL, 2 min), followed by 4% paraformaldehyde in 0.1 M phosphate buffer (PB; pH 7.4, 200 mL, 10 min, 10–12 °C). The brain stem was extracted and a block containing at least the rostral 2/3 of the Sp5C nucleus (including the first cervical spinal segment and the medulla up to the obex; [98]) was removed, postfixed in the same fixative overnight, and subsequently cryoprotected for 2 days in 30% sucrose in PB. Control animals were likewise treated in parallel.

The blocks selected were frozen and serially cut at 40 mm in the coronal plane using a sliding microtome (Leica SM2400, Leica Biosystems, Nussloch). All sections were processed free-floating. Sections were incubated for one night at 4 °C with a combination of three primary antibodies, rabbit anti-Iba1 (1:500; Wako, Richmond, VA, USA), mouse anti-GAD6 (1:50; Hybridoma Bank; Iowa City, IA, USA) and rat anti-GlyT2 (1:100; [99]). After several washes with saline PB (PBS), the sections were incubated for 2 h in the dark in a mixture of secondary antibodies: AlexaFluor 488 donkey-anti-rabbit (green), AlexaFluor 647 goat-anti-mouse (far-red) and AlexaFluor 546 goat-anti-rat (red). In addition, all nuclei were labeled with Bisbenzimide (Hoescht, Thermo Fisher Scientific; Waltham, MA, USA). Sections were then mounted on glass slides, dehydrated, defatted, and coverslipped with Fluoromount-G (Thermo Fisher Scientific, Waltham, MA, USA).

### 4.5. Densitometry of Immunofluorescence

Confocal images covering the rostrocaudal extension of the TCC in both sides were acquired at 1024 × 1024 pixels from 5 sections of each animal with a TCS SP5 Spectral Leica confocal microscope (Leica; Wetzlar, Germany) using a 20 × dry objective. Image stacks and merged channel panels were obtained with Leica LAS AF software (Leica; Wetzlar, Germany). The images obtained were collapsed to create TIFF files with the projections of maximum intensity using the same final thickness of tissue analyzed for all series (5 microns). These images were converted to 8-bit grayscale using ImageJ image analysis software for Windows (Microsoft; Alburquerque, NM, USA). The image processing was identical for all of them, limiting itself to small adjustments in the grayscale and brightness to improve the visualization. Four regions of interest were delineated as broad as possible on the images that selected, respectively, the exterior of the tissue, the white matter forming the outer component of the spinal and medullary dorsal horn, laminae I-II, and lamina III. In each of them, a densitometric analysis of the immunoreactivity of GAD-6, GlyT2 and Iba-1 was performed, obtaining various optical density measurements using the ImageJ ‘Set Measurement’ routine. The data corresponding to the distribution of pixels in the gray scale were also obtained for each of the measurements. The gray values were taken to an Excel sheet where the histograms and corresponding curves were composed. These data were also used for further statistical analysis.

### 4.6. Statistical Analysis

Descriptive statistics (means and SEM or SD) were obtained using the Excel program (Microsoft Office Professional Plus 2010 for Windows 10). For statistical tests, SPSS software (v. 15; IBM, Armonk, North Castle, NY, USA) and GraphPad Prism software (v. 8.0; San Diego, CA, USA) were used. Histograms and graphs were generated with GraphPad Prism software (San Diego, CA, USA) and were retouched to improve image quality using CorelDraw X3 software (Corel; Ottawa, ON, Canada).

The mean densitometric data were analyzed and compared between sides using the Wilcoxon signed rank test using individual sections por pairwise comparisons. Intergroup, sidewise comparisons were assessed by the Mann-Whitney test, using hemispheres as sampling units. The distribution curves of pixels according to their luminance in the 8-bit gray scale were compared with the Kolmogorov-Smirnov two-sample D test [100]. In all cases the statistical significance was established at a *p*-value < 0.05 and was represented in the Figures by asterisks, for intergroup comparisons, or hash symbols, for between-side comparisons (*, #: *p* < 0.05; **, ##: *p* < 0.01).

## Figures and Tables

**Figure 1 ijms-22-04564-f001:**
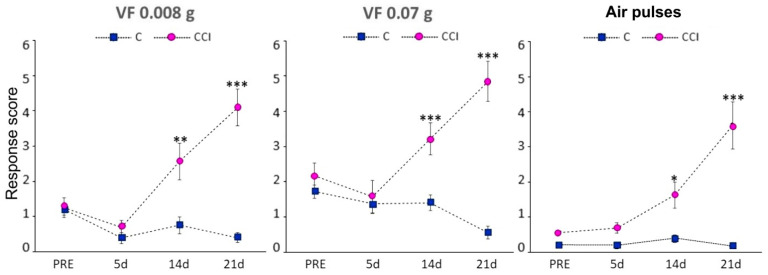
Behavioral responses following CCI-IoN to stimulation of the whisker pad with 0.008 g or 0.07 g Von Frey filaments, or 10 psi air-puff pulses. The response score before surgery (PRE) represents the average of three consecutive testing days. At 14 and 21 postoperative days the response score differed significantly between operated (*n* = 6, right side, magenta circles) and control (*n* = 5, blue squares) cases. The latter included responses from both sides, because there were no left-right differences in these cases. Data represent means ± SEM. Significance is represented by *, ** and ***, corresponding to *p* < 0.05, *p* < 0.01 and *p* < 0.001, respectively (two-tailed Mann-Whitney test).

**Figure 2 ijms-22-04564-f002:**
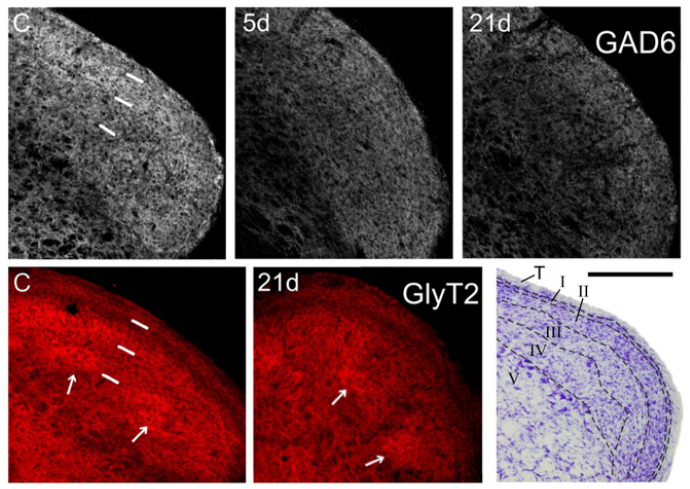
Coronal sections through the right Sp5C caudal to the obex showing immunofluorescence for GAD-6 (**gray, top panels**), and GlyT2 (**red, bottom panels**). GAD-6 expression fades in all laminae at 5 days post-operation (**dpo; 5d**), and decreases even further at 21 dpo, when the loss is more marked in the ventral 1/2 of the nucleus. Less obvious, but still noticeable is a heterogeneous decrease in GlyT2 expression, which affects mostly lamina II and, irregularly, lamina III in the ventral 1/2 of the nucleus (where afferents from the IoN dominate). Arrows point to some ‘patches’ with strong expression of GlyT2 in laminae III-IV, which persist after CCI-IoN. In control sections (**C**) short, white bars indicate the approximate boundaries between the trigeminal tract, laminae I-II, lamina III and lamina IV. These sections were among those used for densitometric measures. A Nissl-stained section (**lower right corner**) shows approximately the same level of Sp5C, with interlaminar boundaries marked by dashed lines. T, trigeminal tract. Scale bar = 500 µm.

**Figure 3 ijms-22-04564-f003:**
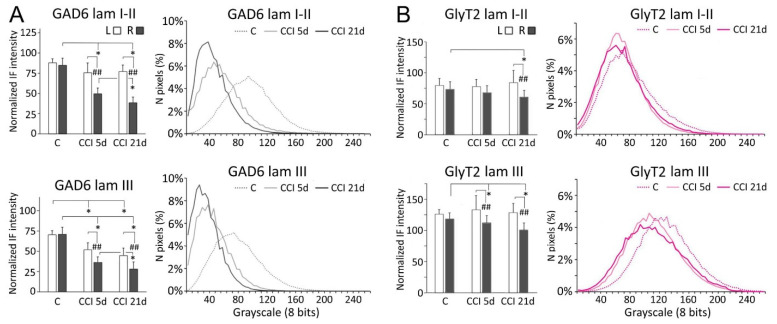
Comparisons of densitometric measurements for GAD-6 (**A**) and GlyT2 (**B**) in laminae I-II (**top panels**) and III (**bottom panels**). Values for each molecule are represented as means ± SD of the normalized fluorescence intensity (histograms), and as the relative distribution of pixels in a grayscale between 0 (darkest) and 255 (brightest). Histograms show values for the contralateral (left, L) and ipsilateral (right, R) sides with respect to the CCI-IoN. Curves show values only for the right side in operated cases, and averages for R and L sides in control cases. C, controls; CCI 5d, operated animals, at 5 dpo; CCI 21d, operated animals, at 21 dpo. In histograms, * indicates statistical significance (*p* < 0.05) between groups, using hemispheres as sampling units (Mann-Whitney test, *n* = 4 in C and *n* = 6 in CCI-IoN cases). Differences between sides in each group were assessed by the Wilcoxon signed rank test using individual sections for pairwise comparisons (*n* = 12 in C and *n* = 18 in CCI-IoN cases; ## = *p* < 0.01). For statistical comparisons of pixel distributions, see Figure 4.

**Figure 4 ijms-22-04564-f004:**
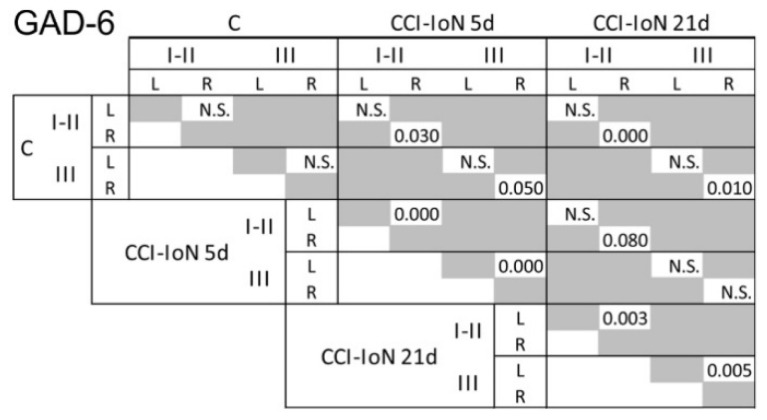
Results of the statistical comparisons (Kolmogorov-Smirnov D test, *p*-values) of densitometric curves for GAD-6 shown in Figure 2. Similar comparisons for GlyT2 did not show any significant difference between groups. N.S., not significant.

**Figure 5 ijms-22-04564-f005:**
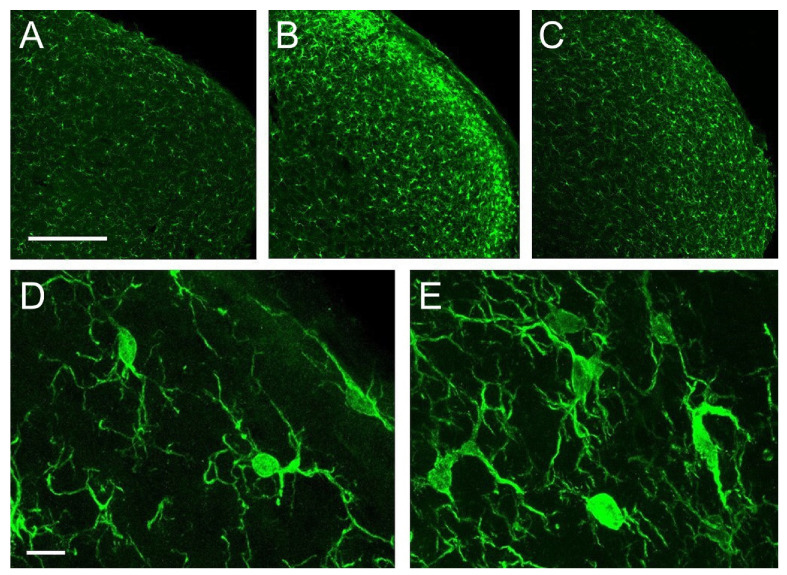
Confocal images of coronal sections of the trigeminocervical complex (TCC) at a caudal level of Sp5C immunolabeled for Iba-1. In controls (**A**,**D**) microglial cells showed a homogeneous distribution, relative low density, and a typical morphology of a ‘resting’ or ‘surveying’ state. CCI-IoN induced a potent microgliosis at 5 dpo (**B**), which decreased but still persisted at 21 dpo (**C**). Activated microglia (**E**) was typically characterized by an increase in the number of cells, which exhibit heterogeneous shapes and sizes, predominating those with larger and irregular somata, and an increased number of processes, including thicker protoplasmic expansions. Scale bars = 500 μm (**A**–**C**) and 10 μm (**D**,**E**).

**Figure 6 ijms-22-04564-f006:**
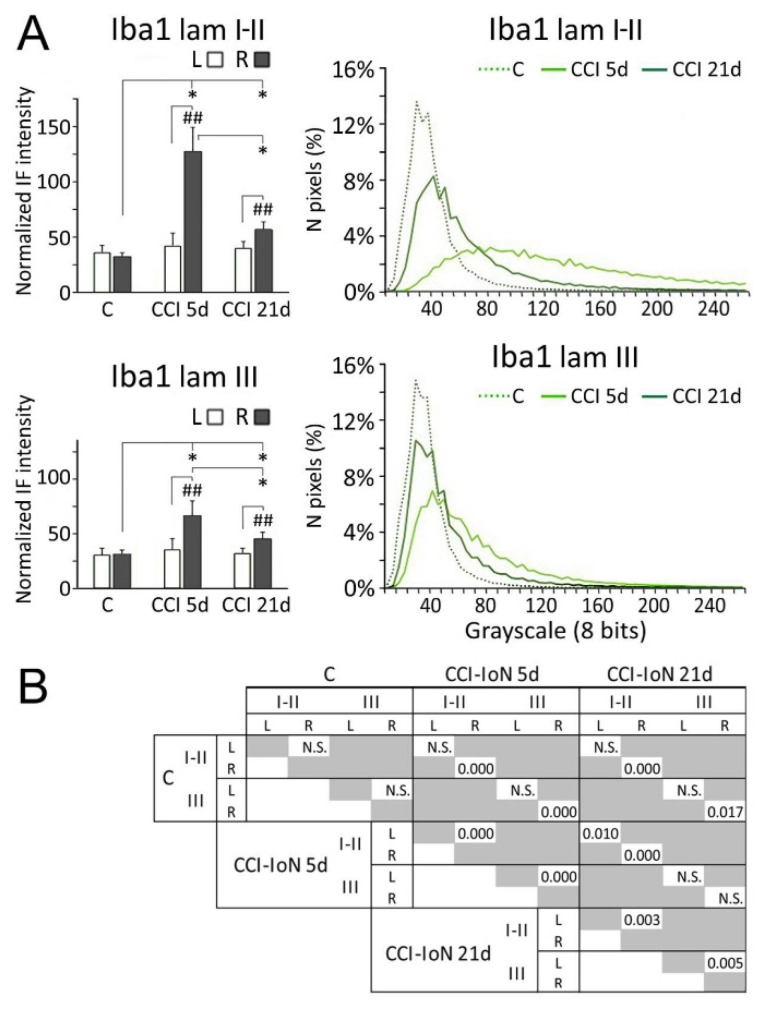
Comparisons of densitometric measurements for Iba-1 (**A**) in laminae I-II (**top panels**) and III (**bottom panels**). In histograms, * indicate statistical significance (*p* < 0.05) between groups, using hemispheres as sampling units (Mann-Whitney test, *n* = 4 in C and *n* = 6 in CCI-IoN cases). Differences between sides in each group were assessed by the Wilcoxon signed rank test using individual sections por pairwise comparisons (*n* = 12 in C and *n* = 18 in CCI-IoN cases; ## = *p* < 0.01). For statistical comparisons of pixel distributions, see Figure 4. The statistical comparisons (Kolmogorov-Smirnov D test) of densitometric curves are shown in (**B**).

**Figure 7 ijms-22-04564-f007:**
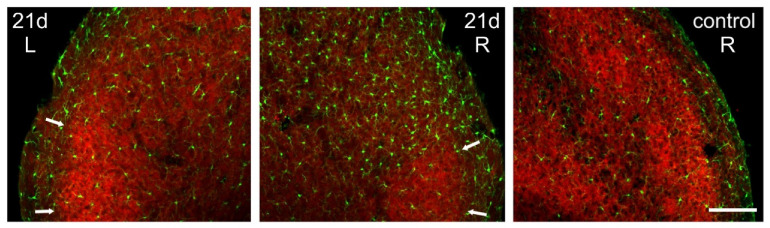
Spatial correlation of microglial reaction to CCI-IoN with GlyT2 expression. The marked ipsilateral microgliosis at 5dpo (not shown) and 21dpo (**right side**, **middle panel**) was practically restricted to the zones of Sp5C exhibiting low expression of GlyT2, sparing the GlyT2-intense ‘patches’ in lamina III (arrows). Scale bar = 300 μm.

**Figure 8 ijms-22-04564-f008:**
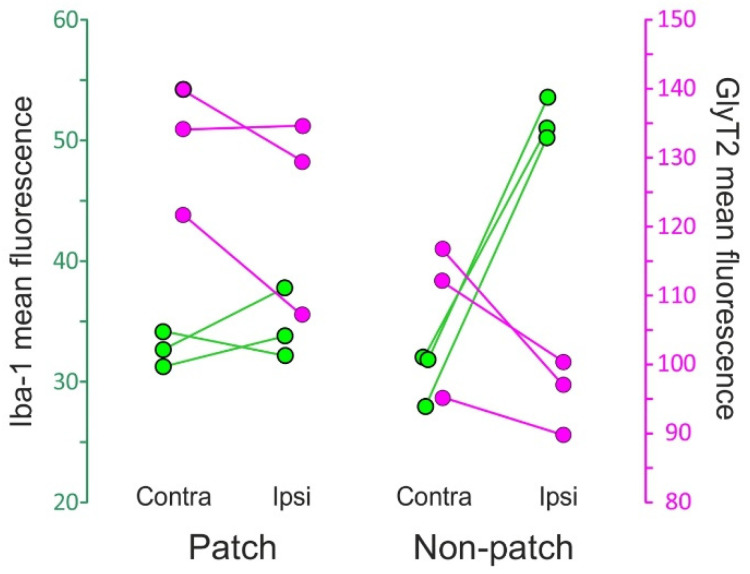
Dot plot of the immunofluorescence of Iba-1 (**green**) and GlyT2 (**magenta**) in lamina III of Sp5C from three CCI-IoN animals. Data represent mean values of 3 sections per case, separately for the GlyT2-strong ‘patches’ and the more moderately immunoreactive ‘non-patch’ zones. Scales are taken from grayscale densitometries with ImageJ (NIH, Bethesda, MD, USA). The diagram shows different combined effects of the nerve injury on each zone: in ‘patches’ the change in Iba-1 immunofluorescence in the side ipsilateral to the CCI is practically nil with respect to the contralateral side, while in ‘non-patch’ or ‘matrix’ zones a pronounced increase is observed. GlyT2, on the other hand, decreases in both zones, apparently more noticeably so in the ‘non-patch’ zones.

**Figure 9 ijms-22-04564-f009:**
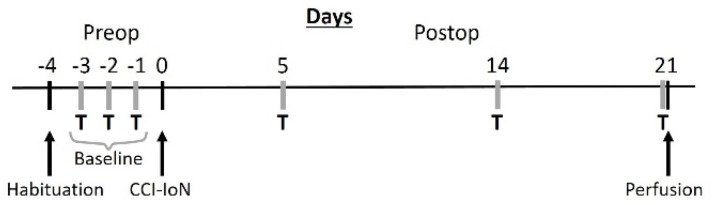
Schematic diagram of the experimental protocol. T, behavioral testing; CCI-IoN, constriction of the IoN.

## Data Availability

The datasets generated for this study are available on request to the corresponding author.

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
