# Peer review of "Microglia and Inhibitory Circuitry in the Medullary Dorsal Horn: Laminar and Time-Dependent Changes in a Trigeminal Model of Neuropathic Pain"

_ijms, 2021, doi:10.3390/ijms22094564_

Round 1
Reviewer 1 Report
The paper has evaluated the changes in microglia activity and inhibitory inputs (GAD-6 and GlyT2 expression) following CCI of the infraorbital nerve (IoN) as a model of facial neuropathic pain in mice. The Authors found that the constriction of the IoN decreased the expression of GAD and GlyT2 with different spatial and chronological patterns in the medullary dorsal horn. This occurred concomitantly with the development of mechanical allodynia and microgliosis. Overall it is a good work, well organized and written, with important evidence for understanding the mechanisms underlying trigeminal neuropathic pain. Minor suggestions are given below
“Among the animal models that mimic chronic pain in the orofacial region, the chronic constriction injury of the infraorbital nerve”….give the abbreviation here.
Write in full when Sp5c TCC, TCS, dpo, are first mentioned, trying to keep the number of abbreviations to a minimum
Page 3 line 82 “Kauf- man [27,28]”. ?
Page 3, lines 84-85: “Following CCI-IoN GAD-6 expression significantly decreased in all laminae of the ipsilateral side by 5 dpo, and this decline continued at least until 21 dpo, even with parallel reductions in the contralateral side compared to controls (Fig. 2).” All this appears interesting but has not been deepened in the discussion
Page 3, lines 89-90 "... and to the contralateral side at 21 dpo were significant in all laminae, but only were significant in lamina III at 5 dpo". This is unclear and should be rewritten
Page 5, line 127 “…a strong microgliosis “ …does it correspond to activated microglia?, morphological changes if observed should be described.
Page 11, lines 331-332: “The experiments started the next day for three consecutive days before IoN surgery, to establish the baseline response score, and on postoperative days 5 and 21”. This is all a bit confusing, perhaps a diagram showing the experimental design and timeline could help the reader. Also, a diagram or illustration of the expression changes in the different laminae could help the reader to appreciate the study.
“The animals were partially restrained in a device that allowed them to expose and move freely the head and forelimbs, so that testing and response recording could easily be made, while producing low stress levels”. Is it a homemade or commercial device?.... if so it should be indicated
Page 18, line 257 “Thereafter, the same effect was found in the dorsal column nuclei after applying local anesthesia to a digit [74,75].” This seems to contradict previous evidence and deserves further discussion
“….spatial disruption of the excitation of inhibitory networks..” it looks a little confusing
Reviewer 2 Report
In this manuscript entitled "Microglia and inhibitory circuitry in the medullary dorsal horn: Laminar and time-dependent changes in a trigeminal model of neuropathic pain", authors aimed at studying the role of GAD and GlyT2 in a CCI model of neuropathic pain at both early (5 dpo) and late (21 dpo) stage of disease. Authors find that CCI reduces inhibitory transmission (i.e. GABA and Glycine mediated) and they also analysed the expression of these markers throughout the time course of neuropathy. Moreover, they found that glycinergic axon terminals patches, highlighted by immunofluorescence analysis, were associated with reduced microglial reactive activation.
I think that the manuscript is well written and pleasant to read. I just have some minor points that need to be addressed:
- Please revise and define all abbreviations (eg Sp5c: superficial trigeminal nucleus caudalis);
- Authors indicate tissue sections as "Coronal sections". They actually look like transverse or axial sections. I would also indicate Rexed laminae with dash lines and numbers (eg PMID: 19723624 PMID: 31030416).
- Please revise the manuscript for some typos and spelling (eg section 2.2 - line 82 "[...] confirming their affinity for axons and terminals (Kaufman [27,28]"
- Authors may find appropriate to speculate and include in the discussion a potential role of astrocytes in modulating or driving microglial phenotype. I also suggest removing references to unpublished observations (Y.B. Martin and C. Avendano).
